# Early-exposure to new sex pheromone blends alters mate preference in female butterflies and in their offspring

Emilie Dion [1]*, Li Xian Pui[1], Katie Weber[1] & Antónia Monteiro [1,2]*

While the diversity of sex pheromone communication systems across insects is well documented, the mechanisms that lead to such diversity are not well understood. Sex pheromones constitute a species-specific system of sexual communication that reinforces interspecific reproductive isolation. When odor blends evolve, the efficacy of male-female communication becomes compromised, unless preference for novel blends also evolves. We explore odor learning as a possible mechanism leading to changes in sex pheromone preferences in the butterfly *Bicyclus anynana*. Our experiments reveal mating patterns suggesting that mating bias for new blends can develop following a short learning experience, and that this maternal experience impacts the mating outcome of offspring without further exposure. We propose that odor learning can be a key factor in the evolution of sex pheromone blend recognition and in chemosensory speciation.

[1] Department of Biological Sciences, National University of Singapore, 14 Science Drive 4, Singapore 117543, Singapore. [2] Yale-NUS-College, 6 College Avenue East, Singapore 138614, Singapore. *email: dion.emilie@ymail.com; antonia.monteiro@nus.edu.sg

The evolution of olfactory sexual communication is a fascinating area of evolutionary biology because changes in pheromones or their perception may lead to assortative mating, reproductive isolation, and eventually speciation. In insects, sex pheromones are critical to the process of finding and selecting a mate[1,2]. The composition and relative proportion of the blend components are species-specific and, together with the corresponding specific receptors, play a fundamental role in interspecific reproductive isolation[3,4]. Recent studies in Lepidoptera support a key role of this chemosensory system in speciation, where pheromone preferences have diversified along with the evolution of the respective blends[5–8]. However, there is still very little understanding of the mechanisms facilitating divergence in mate preferences for new pheromone blends.

Learning to prefer a novel mate signal early in life could be a mechanism driving the evolution of new pheromone communication systems. Learned preferences for novel mate visual signals were previously shown in several arthropods[9]. Early exposure to new ornamentations in spiders[10], fruit flies[11], or butterflies[12] led to shifts in mate preferences in sexually mature older individuals. These premating experiences were thus proposed to play a significant role in reproductive isolation[13,14]. Like visual learning, odor learning happens routinely in an insect's life. For instance, honeybees learn pollen odors while foraging or after being exposed to pollen at an early age[15], and parasitoids learn the odors of their hosts when laying their eggs[16]. Moths can also learn to associate a sex pheromone component with a food reward[17]. To date, however, there is no data on whether any insect can learn to prefer novel pheromone blends. If novel pheromone blends evolve via genetic change, a preference for those blends might first evolve via learning and subsequently become genetically assimilated and fixed in a population.

A mechanism that could accelerate the process of genetic assimilation is the transgenerational inheritance of acquired traits[18]. Behavioral variations following an environmental experience may be caused by epigenetic modifications affecting the expression of relevant genes, which can be inherited through the germline[19]. Inheritance of learning and memory processes has already been shown in several species. Attraction of the nematode *Caenorhabditis elegans* to odors after exposure to these cues was shown to be passed-down to their naïve offspring for several generations[20,21]. In mice, deterrence towards an odor was transmitted to the next generation together with a hypomethylated form of the corresponding receptor gene expressed in the olfactory system[22]. These examples illustrate how learning to avoid or prefer an odor might be transmitted via epigenetic factors to the offspring. If epigenetic modifications, such as silencing marks, alter gene expression for a few generations, shielding these regions from natural selection, then genetic mutations can accumulate in these same regions, eventually stabilizing the phenotype that was originally environmentally induced[23]. To contribute to this field of research, we tested whether female butterflies can shift their mate preferences after being exposed to males with novel sex pheromone blends, and whether learned preferences can be transmitted to the next generation.

We performed these experiments on *Bicyclus anynana* butterflies, which have life-history traits conductive to mate preference learning. Individuals can mate multiply and live up to several months in their natural environment[24]. In cohorts of the same generation, males also usually emerge a few days before females[25]. These traits provide the females with opportunities for adjusting their future sexual behavior based on previous experience with males. In particular, females can learn preferences for novel male wing patterns if they are exposed to them after emergence[12] and if the males express the correct pheromone blend[26]. Newly emerged virgin females frequently reject courting

mates, being exposed to sex pheromones during this process[27,28]. These life history traits make pheromone learning experiments in this species ecologically relevant.

To decide on the type of blend manipulations to do for this experiment we compared the pheromone blend of *B. anynana* with that of closely related species. We also took into account that in insects, new sex pheromone blends may evolve in their composition by loss or gain of single components, or by variation in the ratios of components (e.g. refs. [8,29]). The blend of *B. anynana* contains three male sex pheromone components (MSP) produced after emergence in wing glands[28,30]: (Z)-9-tetradecenol (MSP1), hexadecanal (MSP2), and R6, R10, R14-trimethylpentadecan-2-ol (MSP3). In *B. anynana* of the wet season form, with conspicuous eyespots, females are the choosy sex and males produce high levels of the three pheromone components[30–32]. A comparative study across *Bicyclus* species showed that close relatives vary both quantitatively and qualitatively in their MSP. Sympatric pairs of species display larger differences in component amount and identity than allopatric pairs, suggesting that the blend impacts pre-mating reproductive isolation in this genus[5]. We decided, thus, to vary the amounts of components in our odor learning experiment in *B. anynana*.

We created two New Blends (NB) by either preventing the release of MSP2 and reducing the amounts of MSP1 and MSP3, thus creating a 'reduced' blend (called NB1); and by increasing the amount of MSP2, producing an 'enriched' blend (called NB2). We exposed newly emerged females to these NB and to males with the respective control manipulations (Wild-type—Wt1 or Wt2) and observed the mating outcome of the same females a few days later. We refer below to experiment 1 for all the trials involving NB1 and Wt1; while experiment 2 included all manipulations related to NB2 and Wt2. We tested whether (1) naïve females show mating biases towards either male type; (2) females alter their mating patterns after being exposed to males with NB and; (3) naïve offspring of exposed females show mating patterns different from their mothers. These experiments revealed that mating bias for NB can develop after a short exposure, and that this maternal experience impacts the mating bias of offspring without further exposure. Odor learning may lead to changes in sex pheromone preferences, possibly facilitating chemosensory speciation.

## Results

**Male manipulations altered the levels of MSPs**. We decided to manipulate the levels of MSP2 because males producing higher absolute or relative amounts of this component have a higher mating success[33]. By blocking the pheromone gland on the male hindwing or by perfuming the wing with MSP2, we created two novel blends, NB1 and NB2 respectively, that were different from Wt1 and Wt2 control blends (Fig. 1). These control blends were produced by adding the block solution to the other side of the wing (Wt1), or by perfuming the gland with solvent only (Wt2) (Fig. 1; Supplementary Note 1). MSP2 was absent in NB1 males and was increased by 50-fold in NB2 males (30 min after perfuming) (Figs. 1b, c and Supplementary Fig. 1). Total amounts of MSP1 and MSP3 were reduced by an average of 70% and 60% respectively in NB1 males compared to Wt1 males (Fig. 1b).

**Exposure to new MSP-modified female innate mating bias**. The mating bias of naive females without any social experience was monitored in a mate-choice assay, where the identity of the male (NB1 vs. Wt1; or NB2 vs. Wt2) that mated first with that female was scored. To test if female mating outcome changed after a short social experience, we exposed different females to either NB1, NB2, or to their corresponding control wild type males for a

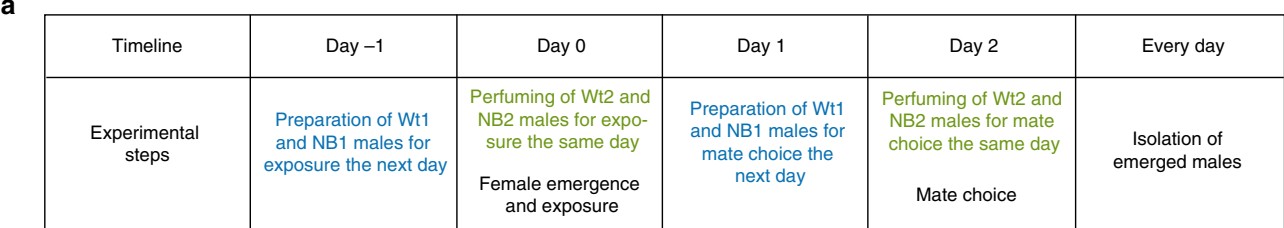

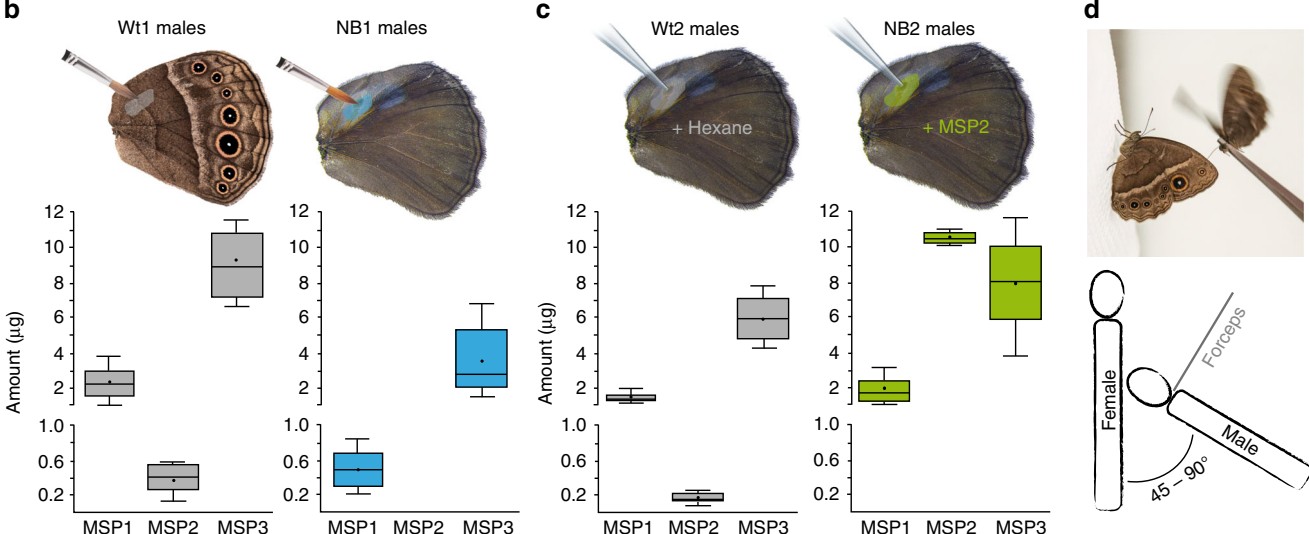

**Fig. 1 Experimental procedure. a** The timeline of the experiment indicating when each step was performed. **b** Coating of the male hindwing androconia (NB1 males) prevented the release of MSP2 and reduced the total amount of MSP1 and MSP3 per male (experiment 1). **c** The average total amount of MSP2 per NB2 male, 30 min after perfuming with synthetic hexadecanal, is increased compared to Wt2 males (experiment 2). In each graph, the horizontal line and the point in each box are the median and the mean amount, respectively. The 25th and 75th percentiles are contained within the outline of the boxes, and the horizontal lines above and below each box show the 1.5 times inter-quartile range of the data. Sample sizes were: $n = 5$ Wt1 males, $n = 7$ NB1 males, $n = 5$ Wt2 males, and $n = 10$ NB2 males, all independent replicates. **d** Schematics of the female exposure where the bottom panel illustrates the position of both male and female individuals from a top view. Source data are provided as a Source Data file.

period of 3 min and scored mating outcome 2 days later. We used mixed models to measure the effects of the age of males used for exposure and for mating trials (from 4 to 6 days old), along with the effect of exposure treatment on female mating outcome, including the female family as a random factor. We also tested female mating biases using Pearson's $\chi^2$ tests (to test for differences from a random mating outcome) (see the "Methods" section for full details).

In experiment 1, naïve females showed an innate mating bias for the wild type blends, with 77% of them mating first with Wt1 males (Pearson's test: $n = 31$, $\chi^2 = 15.16$, $p = 9.89e{-}05$; Fig. 2; Supplementary Table 1a). The female premating exposure treatment significantly affected subsequent mating outcomes (GLMM: $\chi^2_4 = 15.14$, $p = 4.44e{-}03$, Supplementary Table 1b). In particular, 90% of the females pre-exposed to Wt1-males mated with Wt1-males, showing a strong significant mating bias for the Wt1-blend (Pearson's test: $n = 28$, $\chi^2 = 16.7$, $p = 7.12e{-}06$; Supplementary Table 1a), whereas females pre-exposed to NB1-males showed no mating bias, mating randomly with either male (51% mated with Wt1 males; Pearson's test: $n = 31$, $\chi^2 = 0.03$, $p = 0.86$; Supplementary Table 1a; Fig. 2). The mating outcomes of females exposed to Wt1 and NB1 blends were significantly different from each other (Post-hoc tests from GLMM, adjusted $p = 0.018$, Supplementary Table 1c).

In experiment 2, naïve females also had an innate mating bias for the wild type blend, as 70% of them mated with Wt-2 males (Pearson's test: $n = 37$, $\chi^2 = 6.08$, $p = 0.01$; Fig. 3; Supplementary Table 1a). Female early exposure treatment significantly altered

their subsequent mating outcomes (GLMM, $\chi^2_4 = 13.95$, $p = 7.45e{-}03$, Supplementary Table 1b). In particular, females that were pre-exposed to Wt2-males showed no longer a mating bias, with 51% of them accepting the NB2-male first for mating (Pearson's test: $n = 29$, $\chi^2 = 0.03$, $p = 0.85$; Supplementary Table 1a), whereas NB2-exposed females showed a significant mating bias for NB2-males (70%; Pearson's test: $n = 44$, $\chi^2 = 7.36$, $p = 6.66e{-}03$; Supplementary Table 1a). The mating outcomes of naïve females and of those exposed to NB2-males were significantly different from each other (Post-hoc tests from GLMM, adjusted $p = 3.70\ e{-}03$; Supplementary Table 1d).

**Offspring had similar mating outcomes as exposed mothers.** To test for inheritance of learned preferences in experiment 1, we submitted each naïve offspring of NB1-exposed and of Wt1-exposed females to mating trials with a single NB1 and a single Wt1 male. Note that the mothers of these female offspring, despite differences in early odor exposure, were all mated to Wt males in order to remove female choice as a variable in the analysis. Offspring of females exposed to Wt1 males showed a mating bias towards Wt1 males (72%; Pearson's test: $n = 50$, $\chi^2 = 9.68$, $p = 0.002$, Supplementary Table 1a), whereas offspring of females exposed to NB1 males did not show any mating bias, mating randomly with both male types (57% mated with Wt1-males; Pearson's test: $n = 46$, $\chi^2 = 0.78$, $p = 0.38$), as did their mothers (Fig. 2). The percentage of mating with NB1 males was 15% higher in offspring of NB1-exposed females than in offspring

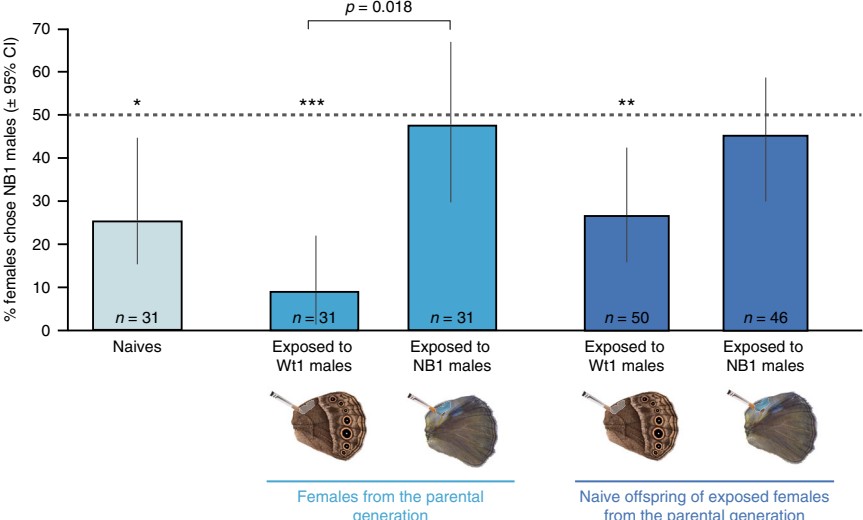

**Fig. 2 Mating outcomes of females after exposure to NB1 and Wt1, and mating outcomes of their offspring.** Mating outcomes shifted after females were exposed to a male with a reduced blend. Most naïve females, and females exposed to Wt1 blend mated with Wt1 males, but females exposed to NB1 males mated with these males at significantly higher rates than Wt1-exposed females. Naïve offspring of females exposed to Wt1 males mated preferentially with Wt1 males, similarly to naïve and Wt1-exposed females from the parental generation. However, offspring of NB1-exposed females mated equally with either male type. Asterisks (*$p < 0.05$; **$p < 0.01$; **$p < 0.001$) indicate statistically significant mating biases for the Wt1 blend using Pearson's $\chi^2$ test. CI means confidence intervals. The dotted lines at 50% illustrates random mating. The horizontal bar above the plot shows a significant difference in mating outcome between the two treatments (from the Tukey post-hoc test, adjusted $p$ value is indicated). The '$n$' on each bar indicates the total number of independent females tested. Post-hoc test results providing adjusted $p$ values comparing the different treatments are shown in Supplementary Table 1a. Source data are provided as a Source Data file.

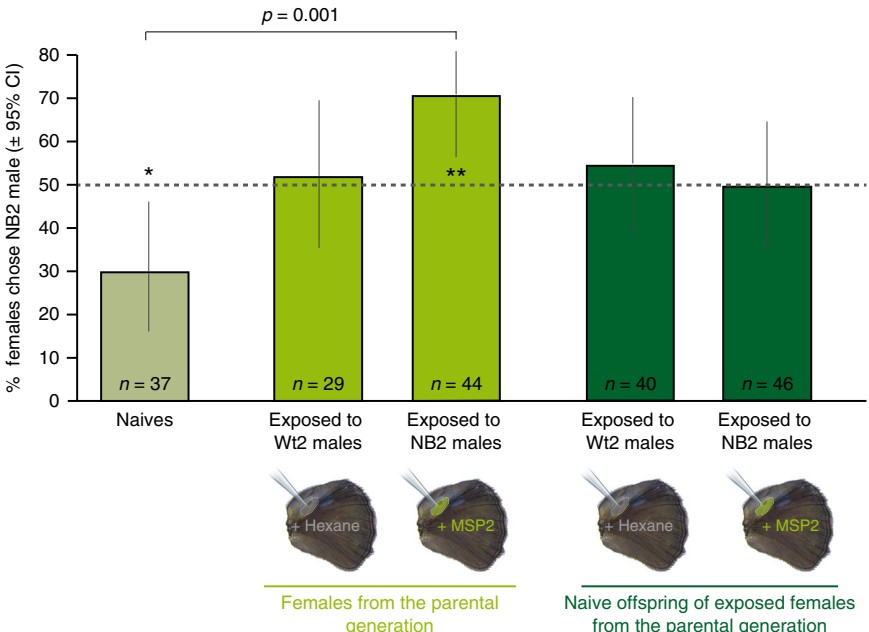

**Fig. 3 Mating outcomes of females after exposure to NB2 and Wt2, and mating outcomes of their offspring.** Females shifted their mating outcomes after exposure to a Wt2 male or a male perfumed with a novel pheromone blend containing more MSP2. Most naïve females mated with Wt2 males, females exposed to the Wt2 blend mated equally with both male types, and females exposed to the NB2 blend mated with NB2 males at significantly higher rates than naïve females. Naïve offspring of females exposed to Wt2 and to NB2 males mated equally with either male type but mated with NB2 males more frequently than naïve butterflies of the previous generation. The dotted lines at 50% illustrates random mating. Asterisks (*$p < 0.05$; **$p < 0.01$) represent significant mating biases tested with Pearson's $\chi^2$ tests. CI means confidence intervals. The horizontal bar above the plot shows a significant difference in mating outcome between the two designated treatments (from the Tukey post-hoc test, adjusted $p$ value is indicated). The '$n$' on top of each bar indicates the total number of independent females tested. Post-hoc test results providing adjusted $p$ values comparing the different treatments are shown in Supplementary Table 1b. Source data are provided as a Source Data file.

of Wt1-exposed females (Fig. 2). For a difference of this magnitude (i.e., effect size) to be significant across offspring types, the sample size would need to be increased to an average of 275 tested female offspring in each group (Supplementary Table 2).

We repeated the same experiment with offspring of Wt2-exposed and NB2-exposed females. Contrary to naïve females from the parental generation who had an innate mating bias for the wild type blend, offspring of Wt2-exposed females mated randomly with Wt2 and NB2 males (55% of them mated with NB2 males) (Pearson's tests; $n = 40$, $\chi^2 = 0.40$, $p = 0.53$) (Fig. 3; Supplementary Table 1a), and had a similar mating outcome as their exposed mothers (Fig. 3; Supplementary Table 1a). While females exposed to NB2 males had a significant mating bias for NB2 blends, their offspring mated randomly (50%; Pearson's tests; $n = 46$, $\chi^2 = 0.00$, $p = 1.00$) (Fig. 3b; Supplementary Table 1a). The mating outcomes of both offspring groups, of their respective mothers and of the females from the parental generation are similar to each other (Supplementary Table 1d).

In all experiments, the age of males used for the pre-mating exposure, mating trial, the position of the black dot placed on the wings to differentiate NB2 and Wt2 males (experiment 2) and the family did not significantly affect mating outcome (Supplementary Table 1b).

## Discussion

The mating outcomes changed after female exposure to mutant pheromone blends. Naïve females mated more frequently with Wt-blend males over males with either of the mutant blends tested. These results demonstrate the ability of the olfactory circuitry to distinguish the different blends and confirm that specific MSP (or their ratios) are important factors determining *B. anynana* mating outcomes[27,28,33]. We demonstrate, however, that an early and brief exposure of females to novel pheromone mutant blends alters their subsequent mating outcomes. Mating outcomes initially biased against NB1-males lacking MSP2 and producing less MSP1 and MSP3 components, lose their bias after a short early-exposure of females to the mutant blend. More strikingly, females mated more frequently with males with high amounts of MSP2, after they were exposed to this new blend, while mating more frequently with Wt-males as naïve individuals. The changes in mating outcome are likely to have resulted from a change in female behavior rather than from alterations in male–male competition or male behavior during the mating trial due to the male's different odors. This is because the mate-choice experimental set-up with both males was identical in every treatment. In addition, the shift in the butterflies' mating outcome was not influenced by mate-choice copying[34], as all females were isolated from each other and from the males since the pupal stage, and visually isolated from each other at every step of the experiment, including during mating trials. These results lead us to suggest that the observed shifts in mating outcome are due to a shift in female preference, and that preference for a male pheromone odor blend in *B. anynana* is not fixed but plastic and influenced by early pheromone odor experiences.

Female preference learning was stronger towards NB2 than NB1, but it is still unclear why this was the case. In particular, NB2-exposed females preferred NB2 over Wt2 males, but NB1-exposed females only lost their preference bias towards Wt1 males, mating randomly with either male type. Previous work showed that males with higher absolute or relative levels of MSP2 to other MSP components had higher mating success[33]. Here, our data for naïve female mating outcome showed that females actually discriminate against males with very high levels of MSP2, but upon exposure to these high levels, females subsequently mate more frequently with these males. We propose that it might be

harder for exposed females to overcome the unattractiveness of NB1 compared to NB2 because NB1 is a highly divergent blend lacking MSP2, whereas NB2 has increased amounts and relative ratios of MSP2. Another possibility for this asymmetry in odor preference learning is that female exposure to enhanced blends (with additional components) relative to Wt blends leads to an overall stronger mate discrimination ability, whereas exposure to weaker blends relative to Wt leads to reduced mate discrimination abilities. When females are exposed to low amounts or absence of components (as was the case for NB1-exposed and Wt2-exposed butterflies), they become less discriminatory and mate randomly with either male. However, when females are exposed to higher levels of blend components (such as for Wt1-exposed and NB2-exposed females), they discriminate better between NB and Wt males, preferring the blend they have been exposed to. We also note that an increase in MSP2 amount alone (as in NB2 males) is sufficient to trigger a change in female discriminating abilities, confirming that this component is important in *B. anynana* mate choice. The neurological mechanisms involved in this process, however, are still unclear.

Alterations of the chemosensory system may be responsible for the change in female blend preference. Brief exposures to odors were previously shown to impact the expression of olfactory receptors, odorant-binding proteins, and the development of brain olfactory centers in honeybees and moths[35–38]. In the bee, qRT-PCR analysis revealed that six floral scent receptors were differentially expressed in the antenna depending on the scent environment they experienced[35]. A brief one-minute exposure of male moths to female sex pheromones led to the up-regulation of a pheromone-binding protein in the male antennae, an enlargement of the antennal lobe, and an increase in the volume of the mushroom bodies in the male brain, which resulted in a higher sensitivity of the exposed males to the female blend[36–38]. The brief exposure of *B. anynana* females to the new male pheromone blend may have led to similar changes in the female brain. The mechanisms in place, however, require future exploration.

Learning to prefer a mutant blend male may have important evolutionary consequences. Mate choice learning has been documented in many insect species[9] but there is still little evidence of the selective advantage of learning sexual traits or preferences, particularly in the wild. Despite this unknown information, both empirical and theoretical studies have highlighted how the learning of a trait or a mate preference can impact assortative mating and population divergence[13,39]. Depending on the specific ecological conditions, type of trait, or learning process, models predict that mate preference learning can lead to reproductive isolation (e.g. refs. [14,40]). Moth and butterfly sex pheromone blends are highly species-specific, ensuring the precise recognition of a compatible mate. These blends are generally thought to be under stabilizing selection because altered signals are less attractive and are thus selected against[41]. However, the learning process that we describe here, by allowing males with divergent blends to reproduce, may mitigate the strength of stabilizing selection, and create opportunities for pheromone blends and reception systems to evolve. A recent study suggested that quantitative and qualitative variations observed in blends within and between natural *B. anynana* populations are potentially catalyzing ongoing speciation[6]. The odor learning ability of *B. anynana* females has probably maintained the high variance in MSP amounts measured in different stock populations[28,30,33], as well as the variance in MSPs detected across natural populations[6]. Furthermore, the use of multimodal signals in mate selection in *B. anynana*, where females use both olfactory and visual cues to assess mate quality[12,26,27], may facilitate pheromone learning and the evolution of the MSP blends. The presence of species-specific visual cues on the male

wings likely increases a female's acceptance of odor-unattractive males from the same species and decreases the risks of females learning NB from hetero-specifics that could lead to hetero-specific mating. Thus, learning to prefer novel odors or odor blends, once these blends arise by genetic mutation and can be inherited by male offspring, may be a key starting point in the process of reproductive isolation and speciation, especially if this learned preference can be transmitted to the next generation via the germ line.

Transgenerational inheritance of pheromone preferences may facilitate the evolution of assortative mating and speciation. Naïve offspring of females exposed to NB1 blends stopped avoiding NB1 blends, as did their mothers. Because these female offspring were not exposed to any blend before the mating assay, we propose that their habituation towards this new blend may have been trans-generationally inherited. Daughters of females exposed to Wt1-blend males, however, did not increase their preference for these males. This lack of transmission of a more extreme preference for Wt blends in female offspring could be explained by an exhaustion of genetic (or epigenetic) variation, since exposure of females to wild type butterflies has been repeatedly done presumably since the origin of this species. Female exposure to NB1 blends, on the other hand, might facilitate the spread (via drift) of NB1 mutant males, if they were to occur for instance at the edges of the distribution of a wild type population.

Naïve female offspring of mothers exposed to both Wt2 and NB2 blends females also mated randomly, contrary to naïve females from the parental generation who preferred the Wt2 blend. However, the preference for blends with higher MSP2 amounts of NB2-exposed females did not translate to a preference for this blend in their offspring, but to a lack of preference instead. We propose that a females' sensitization to MSP2 increases with the amounts of MSP2 she is exposed to, which can happen via exposure to both Wt2 (low levels of MSP2) and NB2 males (high levels of MSP2), but there appears to be a limit to the amount of sensitization that can be transmitted. Different mechanisms of odor preference learning and of odor preference inheritance may explain this phenomenon. It is also possible that several distinct mechanisms lead to changes in female odor preference after exposure, while only one of these mechanisms is involved in the odor learning transmission, leading to a weaker response of naïve offspring. Additional experiments are needed to understand the processes involved. Because all F1 individuals were kept completely isolated from their conspecifics until the mating assay, a change in F1 female preference is also not a result of social transmission, but more likely mediated via epigenetic mechanisms[42].

The transgenerational inheritance of acquired behaviors remains a controversial topic despite the growing number of empirical works supporting it, including mechanistic studies. For instance, first-generation and second-generation naïve *Drosophila melanogaster* offspring displayed a preference toward the alcoholic odors their parent were trained to like. Disruption of the F0 olfactory receptors and specific neuron inputs into the mushroom bodies abolished the change in offspring response, identifying potential targets of epigenetic transmission[43]. In addition, a number of studies have revealed that DNA methylation regulates memory formation and learning processes in insects (e.g. in bees[44,45]) but no study has investigated whether these marks can be inherited to the next generation. Inheritance of a differentially methylated odor receptor gene, however, was shown to take place in mice that learned to avoid a specific odor[22]. This mechanism has not been shown in insects yet, but we may speculate that in our system, genes involved in odor perception and/or processing may have mediated the transmission of odor preferences to female offspring via yet unknown epigenetic mechanisms. A

transmission of acquired pheromone odor preferences may favor assortative mating and chemosensory speciation. Additional experiments and models are needed to understand the mechanisms and the evolutionary consequences of odor-learning inheritance in Lepidoptera.

We have demonstrated the learning and the inheritance of new behavioral responses to new sex pheromone blends by female *B. anynana* butterflies, calling into question the belief that sexual chemical communication is under stabilizing selection. Over time, as new pheromone blends appear, and new learned sex pheromone preferences for those blends develop, new populations of insects may evolve specific sensitivities for those blends encoded at the genetic or epigenetic levels. Learning to prefer a new sex pheromone blend could enable the evolution of chemosensory communication, especially if the learned preferences can be inherited.

## Methods

**Husbandry**. *B. anynana* is an African butterfly that produces alternative seasonal phenotypes in response to environmental cues[46]. To avoid the seasonal variations in courtship behavior[32], eye size and UV light perception[47], and sex pheromone production[30], we performed all experiments with wet season butterflies, all reared at 27 °C, 80% humidity and 12:12 h light:dark photoperiod. Larvae were fed young corn plants, and adults mashed banana. Sex was determined at the pupal stage, and females were placed in individual containers stored in a separated incubator, devoid of males or male sex pheromones until a male exposure or a mating trial. Upon emergence, males were put in age-specific cages. Males were all naïve, virgin, aged from 4 to 6 days old during the experiment, reared in the same conditions, and had dorsal forewing eyespot UV-reflective pupils (as their absence in males is strongly selected against by females[31]). The two males presented to each female for a mating trial had the same age and similar wing size. In both experiments, different males were used for exposure and mating trials. The experimental procedure is described in Fig. 1.

**Experiment 1: Prevention of MSP2 release from males**. In this species, the three MSP components are produced by the hindwing gland, while two components, MSP1 and MSP3, are produced by the forewing gland (Wt1 and Wt2 males in Fig. 1b, c)[28,30]. NB1 males were prepared following the method described in ref. [27]. The ventral hindwing androconia and yellow hair pencils were both coated with transparent non-viscous nail solution (Revlon Liquid Quick Dry #990). The hindwing dark hair patch, which overlaps the forewing androconia, was left uncoated. This treatment prevents the emission of MSP2 produced by hindwing glands only[28], and causes the reduction of MSP1 and MSP3 total amounts by an average of 70% and 60%, respectively (Fig. 1b)[30]. The hindwing ventral side of Wt1 males received the same treatment to control for the odor of the nail solution. NB1 and Wt1 males were prepared ~16 h prior to exposure or mating trials (Fig. 1a, b).

**Experiment 2: Increase of MSP2 amount in males**. 5 μg of MSP2 (≥98% purity, Cayman Chemical, no. 9001996) diluted in 2 μL of hexane (99% purity, Merck, no. 104372) were applied to each hindwing androconia of NB2 males. Wild type control males (Wt2) received the same volume of solvent only in the same wing location (Fig. 1c). Hexane was used as a solvent as it did not impact naïve female choice (tested in a mating trial, described in Supplementary Note 1). The high load of synthetic hexadecanal was chosen to maximize the difference between MSP2 amounts of NB2 and Wt2 butterflies until several hours after application of the solution (Supplementary Fig. 1). Males were allowed to rest 30 min after perfuming until used for exposure or mating trials. At the end of this 30 min period, NB2 males had similar amounts of MSP2 and MSP3 on their wings (Fig. 1c).

The amount of MSP and the evaporation rate of hexadecanal was determined by gas chromatography from 30 min to 8 h after perfuming on a different set of males. Between perfuming and MSP extraction, two males were placed together in one cylindrical hanging net cage, under identical temperature, humidity and light conditions than the ones used for the mating trials (see section 'Mating trials' below). The MSP extraction procedure and quantification are described in Supplementary Methods 1.

**Female exposure to new blend or wild type males**. The female butterfly was released in a cylindrical hanging net cage (30 cm diameter, 40 cm height) less than an hour after emergence (on day 0). The exposure was done manually by retaining the male between the head and the thorax with narrow-tipped feath-erweight forceps for 3 min. The males were presented directly to the females in a similar way as the natural courtship behavior (same distance and orientation). We encouraged the male to flutter its wings towards the female by gently squeezing the forceps. This process mimicked the first step of the courtship

sequence, the male fluttering, which facilitates the volatilization of the pheromone towards the female (Fig. 1d). This procedure allowed a direct and controlled exposure of the females and was non-harmful to the males. After exposure, the female remained isolated until day 2, when mate choice assays were conducted (Fig. 1a). Each female (naïve included) was allocated an identification number, which does not indicate the treatment she was submitted to, so that the investigator was unaware of the sample group allocation during the mating trials and when assessing its outcome.

**Mating trials**. All experiments were done at 24 °C, 60% humidity, under UV and white light, in cylindrical hanging net cages. Mating trials of naïve and exposed females was started on day 2, around 9.30 a.m. (Fig. 1a). One Wt and one NB male were placed in the same cage along with the female. Female's abdomens were pre-dusted with fluorescent orange powder, which is transmitted to the male upon copulation, allowing the identification of the mating partner. Males were checked for presence of powder every 2 h to prevent multiple mating. Assays were ended 6 h after the beginning of the experiment. The latter time point corresponds to MSP2 amounts becoming similar between Wt2 and perfumed males (Supplementary Fig. 1; MSP2 amounts were on average higher in NB2 males relative to Wt2 males at 6 and 8 h after perfuming. This was significantly so for 8 h males and marginally not significant for 6 h males). To differentiate NB2 and Wt2 males, a black dot was applied with a sharpie pen randomly at the top or the bottom of their ventral hindwing. NB1 and Wt1 males were recognizable thanks to the light gray color of the nail solution covering the androconia or the corresponding area of the wing on the opposite side.

**Testing the inheritance of mate preferences**. An additional group of females were exposed to either NB1, Wt1, NB2, and Wt2 males, following the same exposure protocol as described above. We did not test the preference of offspring of females that chose NB1 and NB2 males, but instead, each exposed female was mated with a single naïve Wt males in a separate cage. This procedure was followed to prevent possible confounding effects of the mating trials and any predisposed genetic preferences that females may have. The male was removed after mating and the female given a corn plant for egg collection. Each female and its offspring (F1 individuals) constituted a family. F1 pupae were sexed, and the females were submitted to the exact same isolation procedure as naïve females until mating trials between a NB1 and a Wt1 male, or between a NB2 and a Wt2 male, tested on day 2, using identical procedures as described above (Fig. 1). From 2 to 8 female offspring per family were tested from the 13 Wt1, the 11 NB1, the 9 NB2, and the 9 Wt2 families.

**Statistical analyses**. A Generalized Linear Mixed Model (GLMM) was used to analyze the effect of the different treatments on females and their offspring mating outcome, as this model includes both fixed and random effects. *Female mating outcome* was the binomial response (NB male chosen or not, coded 1 and 0, respectively). The *family* identity was implemented as a random factor in the models. Each female from the parental generation, taken from our stock population cage, was considered as belonging to different families. The fixed factors included the female *treatment* (NB-exposed females, offspring of NB-exposed females, Wt-exposed females, offspring of Wt-exposed females, and naïve females), the *age of males used for mate choice* (4, 5, or 6 days old) and the position of the black mark used to differentiate males perfumed with NB2 to males perfumed with the solvent only (at the top or the bottom of the ventral side of the left hindwing; in experiment 2 only). Because naïve females and the offspring females were not exposed, the effect of *male age during exposure* (4, 5, or 6 days old) on female choice was analyzed separately with a binomial logistic regression using the parental generation only. *p*-values of factor effect were obtained by likelihood ratio tests of full regression models tested against simplified models with specific factors removed. Adjusted *p*-values were obtained using pairwise comparison of the significant fixed effects using Tukey Contrasts. All measurements were taken from distinct samples. Results from experiments 1 and 2 were analyzed separately using R v. 3.2.4[48] implemented in RStudio v.1.0.136[49]. Packages lme4[50], multcomp[51], car[52], Rmisc[53], and rcompanion[54] were used.

In both experiments, we tested if females had a significant mating bias for the NB or the WT blend using a Pearson's $\chi^2$ test in R. The mating outcome was considered a mating bias if it significantly differed from a random mating (50:50). Each group was analyzed separately, and the analysis output provided the 95% confidence interval displayed in Figs. 2 and 3.

**Compliance with ethical standards**. No ethical approval was needed to work with *B. anynana* wild type laboratory strains.

**Reporting summary**. Further information on research design is available in the Nature Research Reporting Summary linked to this article.

## Data availability

All the data generated during the current study are available on the Institutional repository of the National University of Singapore ScholarBank@NUS (https://doi.org/10.25540/8Y6E-XPMY). The data underlying the figures in the main text and in the Supplementary Information are provided as a Source Data file.

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

## Acknowledgements

We thank the National University of Singapore Environmental Research Institute and Frances Lim for the use of the GC-QQQ; Jeremy Pang and Tan Min for performing preliminary experiments; The Fireflies farm for providing corn plants, Dr. Erica Westerman, Dr. Marie-Jeanne Holveck, Dr. Adam Claridge-Chang and the butterfly lab members for their help and useful suggestions about the experiments, the analysis and the manuscript. This work was funded by the Ministry of Education, Singapore grant MOE2014-T2-1-146.

## Author contributions

E.D. and A.M. designed the study; E.D., L.X.P., and K.W. performed the experiments, E.D. analyzed the data; E.D. and A.M. wrote the manuscript and provided revisions.

## Competing interests

The authors declare no competing interests.
