## [Peer Review File · Nature Communications]

Reviewers' Comments:

Reviewer #1:

Remarks to the Author:

This paper describes interesting learning experiments with the butterfly *Bicyclus anynana* to show that pre-exposure to pheromone affects female choice, even in the second generation. Overall, I think this is a really interesting paper. However, I hope that Caroline Nieberding has also been consulted as reviewer for this paper, as she is the expert on the sex pheromone communication system in this species. In addition, I have the following questions and comments that I hope the authors can address before publication:

1. First of all, I think it is important to point out that *B. anynana* is different from other moth species in that it is relatively long-lived, which makes learning a likely strategy. This is in contrast to most other moth species that generally live only a few weeks.
2. line 89: the wet season form: explain this form for outsider audience; with or without eyespots?
3. line 104: immature females: please explain what you mean with this. How long after hatching are females of *B. anynana* immature and when are they mature? The exposure experiments are done with females directly after hatching, at day 0, so does this mean that these females are still immature? At what age do females start to mate?
4. line 112 "MSP2 was previously suggested to be the most relevant pheromone to female choice"; this is a very strange sentence, as one component of the pheromone blend is not the pheromone of the species. In general, it is not one component, but the ratio between components that is important
5. lines 278-279: "This treatment prevents the emission of MSP2 produced by hindwing glands only". So where are MSP1 and MSP3 emitted from in addition to the hindwing glands?
6. line 284: 5 ug of MSP2 was applied; this is a huge amount, as the authors also write. What was the purity of this compound? Please specify as much as possible, as it is possible that small 'contaminants' may have an effect as well.
7. I can't figure out how long after the perfuming of the male the mate choice experiments took place. Were the same perfumed males used in the female exposure assays and the mate choice assays? In that case, males were perfumed and exposed to females on day 0 and then used in the mate choice trials on day 2? If so, did the authors check the amount of pheromone (MSP1, 2 and 3) on day 2, i.e. right before and/or after the mate choice trials?
8. The females were pre-exposed to males. Were the males that were used for these pre-exposure experiments used also in the mate choice tests or not? If so, then could this pre-exposure have influenced the males in the mate choice trials?
9. line 313: "Males were checked for presence of powder every 2 hours to prevent multiple mating." How long does one mating last? Why were matings not observed e.g. every 30 min?
10. I wonder whether the term 'choosing' is completely correct throughout the manuscript. The authors used mating as the proxy, which may have been due to female choice and/or to interactions between the (competing) males.

Minor comments:

- line 86: sex pheromone is singular; the sex pheromone of a species consists of a number of sex pheromone components (when behavioral response has been established) or compounds (when behavioral significance is not clear or not proven)
- line 94: the authors write "we investigated how pheromone blends vary across closely related species", but this ms deals only with *B. anynana*, so this should be changed.
- line 322: choose should chose, as it is the past tense.

Hopefully these comments are helpful.
Kind regards, Astrid Groot

Reviewer #2:

Remarks to the Author:

COMMENTS TO THE AUTHOR

This is a well written, nicely organized and presented study. Their experimental manipulations of natural pheromone blends are elegant. I think this component of the manuscript is strong. The possibility of generational transfer of preference for learned pheromone blends is intriguing, but I am not convinced that the evidence here is convincing enough for publication.

I am concerned that it looks as if the females in the second generation did not behave significantly differently from each other or the naïve females in the first generation. I think that given that the transgenerational learned preference is a fairly extraordinary claim, it needs incontrovertible evidence, and I am not sure you have that here. I would like to see the preferences of these females compared to the preferences of naïve females, or even better a control group where the mothers had not been exposed to any male pheromones other than the Wt they mated with.

MINOR COMMENTS

Lines 26 – 27: the statement that females are able to "pass that learned preference down to the next generation" needs more clarification and explanation if it is to be included in the abstract.

Lines 111-118: My impression from this section is that NB1 males had overall reduced pheromone amounts (including no MSP2 and 70% and 60% of MSP1 and MSP3). NB2 males on the other hand just had more MSP2, which has already been shown to be more attractive to females (lines 112-113).

Please add an explanation of what the Wt1 and Wt2 control blends are in the text

Line 129: "these variable" do you mean male age? Clarify

Lines 111-154: Please list your sample sizes for each experiment throughout the results (especially since you mention effect size in line 151).

Lines 301-302: sentence is awkward – please reword

Reviewer #3:

Remarks to the Author:

It seems to me that the phenotype of the F1 sons (brothers to the daughters with inherited preferences) is critical to this story. If they do not possess off-ratio blends to mirror the inherited preferences of their sisters, it seems that the alternate outcomes of introgression or extinction are more likely.

It is also not clear to me why learning would alter female preference functions. What is the selective advantage?

I do not like the bioassay. The dose (line 284 = 5 micrograms) seems very high. I realize this was necessary to try to have altered male phenotypes throughout the bioassay period, which is another problem. I question the significance of data generated by an 8-hour endpoint bioassay in a very small arena where females cannot escape male attention and none of the participants can leave. In nature it's quite improbable that males and females would remain in such close proximity for 8 hours. Why didn't the authors observe the interactions? The temporal elements associated with this bioassay (how often, when in the 8 hour window, etc.) seem more relevant than the outcome after such a long period. I also think it is problematic that the modified male phenotype is dynamic during the bioassay interval, returning to wild type phenotype by the end of the 8 hour interval (if I read the MS correctly), particularly since it was an endpoint bioassay and the mating could have happened near the end of the 8 hour period, raising questions about what the preference of these females really is.

Specific

Line 50 – I don't like the term "originating." Perhaps facilitating would be better?

Line 86 – there is one pheromone and three pheromone components.

Line 95 – I do not see this citation (29) as a good supporting reference for this statement.

Line 99 – I think it would have been more meaningful if the variation reflected intraspecific variation rather than variation among congener a novels.

Overall the authors have proposed a novel series of mechanisms that would in their view facilitate the rapid evolution of new pheromone communication systems (at least in butterflies that use a strongly pheromone-biased system). It is incumbent of them to provide definitive proof of the feasibility of their model. For the reasons above, publication ought to await further confirmatory experiments that support their proposed step-wise mechanisms and correct the bioassay's deficiencies. Furthermore, the authors should recognize that although their tests achieved statistical significance, a change in the behavior of a relative small number of butterflies would have eliminated any significance by chi-squared tests.

Reviewers' comments:

Reviewer #1 (Remarks to the Author):

This paper describes interesting learning experiments with the butterfly *Bicyclus anynana* to show that pre-exposure to pheromone affects female choice, even in the second generation. Overall, I think this is a really interesting paper. However, I hope that Caroline Nieberding has also been consulted as reviewer for this paper, as she is the expert on the sex pheromone communication system in this species. In addition, I have the following questions and comments that I hope the authors can address before publication:

Thank you for your comments and suggestions. We have answered all of them point by point below and have edited the manuscript accordingly. We also originally suggested to the editors that Caroline Nieberding review this manuscript.

1. First of all, I think it is important to point out that *B. anynana* is different from other moth species in that it is relatively long-lived, which makes learning a likely strategy. This is in contrast to most other moth species that generally live only a few weeks.

Yes, that is true, thanks for pointing this out. We have added this point to the introduction (L. 88-91).

2. line 89: the wet season form: explain this form for outsider audience; with or without eyespots?
We have added some details L. 104.

3. line 104: immature females: please explain what you mean with this. How long after hatching are females of *B. anynana* immature and when are they mature? The exposure experiments are done with females directly after hatching, at day 0, so does this mean that these females are still immature? At what age do females start to mate?

By immature, we meant that females typically reject courting males until they are 2 days old. This is a behavioral observation, but no information is available about the development of the ovaries. We exposed the females upon emergence (day 0), when they still usually reject courting males, and submitted them to mate choice at day 2, when they usually start mating. This natural ability of females to be courted while refusing to mate with a male, mimics our experimental set-up. We have explained this in L. 94-95. To remove ambiguities, we changed the word "immature" to "newly emerged" (L. 114).

4. line 112 "MSP2 was previously suggested to be the most relevant pheromone to female choice"; this is a very strange sentence, as one component of the pheromone blend is not the pheromone of the species. In general, it is not one component, but the ratio between components that is important

We have removed this sentence and explained that we decided to manipulate MSP2 level because males with higher absolute or relative amounts (ratio) of MSP2 were shown to have a higher mating success (L 123-124).

5. lines 278-279: “This treatment prevents the emission of MSP2 produced by hindwing glands only”. So where are MSP1 and MSP3 emitted from in addition to the hindwing glands?

MSP1 and MSP3 are produced by the forewing gland, while the three MSPs are produced by the hindwing glands. We added this information in the methods section (L. 335-336).

6. line 284: 5 ug of MSP2 was applied; this is a huge amount, as the authors also write. What was the purity of this compound? Please specify as much as possible, as it is possible that small ‘contaminants’ may have an effect as well.

According to the manufacturers, the purity of the synthetic hexadecanal we used is $\geq 98\%$ and the purity of hexane is 99%. We have added this information to the methods section (L. 346).

7. I can't figure out how long after the perfuming of the male the mate choice experiments took place. Were the same perfumed males used in the female exposure assays and the mate choice assays? In that case, males were perfumed and exposed to females on day 0 and then used in the mate choice trials on day 2? If so, did the authors check the amount of pheromone (MSP1, 2 and 3) on day 2, i.e. right before and/or after the mate choice trials?

Different males were used for the female exposure and for the mating trials. Males were perfumed 30 minutes both before being exposed to the females (for 3 minutes) and before the start of each mating trial.

The amount of MSP in perfumed males was calculated from individuals that were not used in any further experiments. All males (the ones used for female exposure, for mate choice, or for MSP amount calculation) had the same age, same experience, were reared the same way, underwent similar manipulation and were placed in the same cage in the same conditions (see Suppl. Method 1). We thus assumed that all perfumed males had similar amounts of MSP. We have clarified these points in the methods section (L. 332, 355-357, Supp. methods 1).

8. The females were pre-exposed to males. Were the males that were used for these pre-exposure experiments used also in the mate choice tests or not? If so, then could this pre-exposure have influenced the males in the mate choice trials?

Different males were used to expose females and in the mate choice trials. This way, we controlled for any male previous experience. We have clarified this point in the methods section.

9. line 313: “Males were checked for presence of powder every 2 hours to prevent multiple mating.” How long does one mating last? Why were matings not observed e.g. every 30 min?

One mating usually lasts an average of 30 minutes. The rate of multiple mating was low (8% in experiment 1 and 5% in experiment 2), and the latency to mating was very variable (from 30 minutes to a few hours). Based on these observations, we established that a 2-hour interval between observations would be suitable to prevent multiple matings without disturbing the experiment and stressing the butterflies too often. This disturbance occurs because the males need to be individually handled and checked for the presence of orange powder on their abdomen that was transferred from the female abdomen. Presence of the powder indicates that the male has mated.

10. I wonder whether the term 'choosing' is completely correct throughout the manuscript. The authors used mating as the proxy, which may have been due to female choice and/or to interactions between the (competing) males.

Yes, that is correct, we didn't measure male behavior components specifically, so we cannot fully exclude the effect of male-male competition on mating outcomes. We have thus replaced "female choice" and "female preference" by "mating outcome" or "mating bias" throughout the manuscript. However, since female experience was the only factor that differed between the treatments, this suggests that the change of mating outcome is more likely attributed to female behavior rather than to male behavior. We have also discussed this point L. 198-199 and started using "female preference" only thereafter in the discussion section of the manuscript (from L. 204).

Minor comments:

-line 86: sex pheromone is singular; the sex pheromone of a species consists of a number of sex pheromone components (when behavioral response has been established) or compounds (when behavioral significance is not clear or not proven)

We have added "components" to this sentence.

-line 94: the authors write "we investigated how pheromone blends vary across closely related species', but this ms deals only with *B. anynana*, so this should be changed.

We have clarified that we researched the literature for this information (now L. 97-100).

-line 322: choose should chose, as it is the past tense.

We have corrected this typo.

Hopefully these comments are helpful.

Kind regards, Astrid Groot

Reviewer #2 (Remarks to the Author):

COMMENTS TO THE AUTHOR

This is a well written, nicely organized and presented study. Their experimental manipulations of natural pheromone blends are elegant. I think this component of the manuscript is strong. The possibility of generational transfer of preference for learned pheromone blends is intriguing, but I am not convinced that the evidence here is convincing enough for publication.

I am concerned that it looks as if the females in the second generation did not behave significantly differently from each other or the naïve females in the first generation. I think that given that the transgenerational learned preference is a fairly extraordinary claim, it needs incontrovertible evidence, and I am not sure you have that here. I would like to see the preferences of these females compared to

the preferences of naïve females, or even better a control group where the mothers had not been exposed to any male pheromones other than the Wt they mated with.

We agree that the inheritance of a learned behavior is a fairly extraordinary claim that requires strong evidence, however, we believe we have provided the tests suggested above as evidence for our claims. In addition, we have made our manuscript more comprehensive by adding a second experiment, examining the naïve mate choices of females exposed to the second enhanced pheromone blend (NB2), as well as those exposed to Wt2 blends, to further strengthen our claims.

As suggested we have compared the preference (mating bias) of naïve females from the parental generation to the preference (mating bias) of their offspring. In experiment1, naïve females from the parental generation and offspring of Wt1-exposed females prefer the Wt blend, while offspring of NB1-exposed females don't show any preference. In experiment 2, offspring of both Wt2- and NB2-exposed females don't show a mating bias, contrary to naïve females from the parental generation who actually have a mating bias for the Wt blend. We have described these results from L. 157 and in Figs 2, 3, and Table 1a). We didn't test the offspring of naïve females that mated with Wt males because we assumed that these offspring have similar mating outcomes to naïve individuals, that we had already tested.

Even if there is no understanding of what causes a change in the behavior of the offspring of exposed females at the mechanistic level, the behavioral change is still shown. We believe that our behavioral data should be published so that other teams can also repeat the experiment and provide additional information about the inheritance of acquired traits. However, since we cannot provide a mechanism in this manuscript we have explicitly stated that understanding these results requires more experiments (e.g, L. 290; 308-309).

MINOR COMMENTS

Lines 26 – 27: the statement that females are able to “pass that learned preference down to the next generation” needs more clarification and explanation if it is to be included in the abstract.

We have clarified this point by describing the actual results in the abstract (L. 25).

Lines 111-118: My impression from this section is that NB1 males had overall reduced pheromone amounts (including no MSP2 and 70% and 60% of MSP1 and MSP3). NB2 males on the other hand just had more MSP2, which has already been shown to be more attractive to females (lines 112-113). Please add an explanation of what the Wt1 and Wt2 control blends are in the text

We have added this explanation (L. 127-128)

Line 129: “these variable” do you mean male age? Clarify

Yes, we mean male age. We changed the text accordingly (now L. 139).

Lines 111-154: Please list your sample sizes for each experiment throughout the results (especially since you mention effect size in line 151).

We have added sample sizes in the text with the corresponding analysis results. They are also displayed on each figure at the top of each bar, and in Suppl. Table 1a.

Lines 301-302: sentence is awkward – please reword

We have changes this sentence (now L. 364-366)

Reviewer #3 (Remarks to the Author):

It seems to me that the phenotype of the F1 sons (brothers to the daughters with inherited preferences) is critical to this story. If they do not possess off-ratio blends to mirror the inherited preferences of their sisters, it seems that the alternate outcomes of introgression or extinction are more likely.

Thank you for your comments. Obviously in this experiment we are merely simulating the origin of a novel blend in males of a population by manipulating the blend artificially. The male offspring of these manipulated males will have wildtype blends. We assume, then, that you are referring to adding the male side of the story in the discussion of our results. If “novel blend” males in a population have evolved these novel blends due to genetic mutations then their male offspring will also have the novel blend as they will have inherited these same mutations. We have clarified this scenario now at two points in the manuscript (L. 65 and 263-264). This is the scenario that we have tested here. Exploring the evolutionary consequences of this type of scenario, however, may require a detailed model of the inheritance of genetic changes affecting the pheromone odor and epigenetic changes affecting the female preference. This can be attempted in future by labs with the relevant modeling expertise.

It is also not clear to me why learning would alter female preference functions. What is the selective advantage?

Mate choice learning has been documented in many insect species but there is still little evidence that learning sexual traits or preference affects the fitness of the individuals that learn these preferences, particularly in the wild. Despite this unknown information, theoretical models (based usually on vertebrate data) show that preference learning may lead to sexual isolation and speciation in specific conditions (reviewed in Verzijden et al. *TREE*, 2012 and in Dion et al. *Frontiers Ecol. Evol.*, 2019). Models have also shown that a preference learning gene can become fixed in a population even if it has no direct fitness benefit (Servedio & Kirpatrick, *Am. Nat.*, 1996).

Here, learning to prefer a new blend may be advantageous when it lowers mate searching costs in isolated populations where mutant blends are frequent. But a precise answer to your question will require additional experiments. We have added these points in the text (L. .244-244; 308-309).

I do not like the bioassay. The dose (line 284 = 5 micrograms) seems very high. I realize this was necessary to try to have altered male phenotypes throughout the bioassay period, which is another problem. I question the significance of data generated by an 8-hour endpoint bioassay in a very small arena where females cannot escape male attention and none of the participants can leave. In nature it's quite improbable that males and females would remain in such close proximity for 8 hours. Why didn't

the authors observe the interactions? The temporal elements associated with this bioassay (how often, when in the 8 hour window, etc.) seem more relevant than the outcome after such a long period. I also think it is problematic that the modified male phenotype is dynamic during the bioassay interval, returning to wild type phenotype by the end of the 8 hour interval (if I read the MS correctly), particularly since it was an endpoint bioassay and the mating could have happened near the end of the 8 hour period, raising questions about what the preference of these females really is.

We are aware that experimental set-ups don't always match environmental conditions, making the outcome of lab experiments less predictable in natural populations. However, this set-up actually allowed us to control for all unpredictable factors and clearly showed that *B. anynana* females can learn new pheromone preferences via a very short 3-minute pre-exposure to novel pheromone blends. This 3-minute exposure is the key point in the experiment and we think our behavioral manipulation it is not too far-fetched as male butterflies often court and chase the same female for this amount of time in our lab populations. In nature, the "mutant blend male" would be a genetic mutant that presumably would not have any attenuation of its pheromone blend over time. So, our experimental set-up, with declining odor titers over time, is only a poor approximation of what would actually happen in nature. In addition, in the field, these butterflies actually stay close to each other for some significant amount of time, especially when they gather on the same feeding spot. Females also are exposed to courting males during their two first days of life without accepting mating, which is similar to the set-up we have used.

Also, the "8 hour" limit was actually a typo. We have made this correction in the text of the manuscript. Displayed data and statistical analysis are from experiments that actually lasted a maximum of 6 hours, so we ensured that NB2 males still had higher MSP2 levels than Wt2 males by the end of the observation period. Note in sup. Figure 1 that Msp2 levels are on average around 2500ng in NB2 males after 6 hrs, while the levels are only around 450ng in Wt2 males.

We agree that recording the behaviors during this 6-hours period could have been interesting, but it was technically challenging, especially as butterflies can remain inactive for hours. Perfuming the NB2 males again during the assay was also not done, as it stresses out the male, and would disturb the experiment. However, we actually have recorded the time taken to mating. and the frequency of matings at each recorded time (2h; 4h and 6h). A similar frequency was observed across treatments (Naives: 14%; 27% and 59%, NB2-exposed: 17%, 30% and 53% and Wt2-exposed: 13%, 32% and 54%, respectively for 2, 4 and 6h). So, mating latency is unlikely to have affected mating outcome between treatments.

Specific

Line 50 – I don't like the term "originating." Perhaps facilitating would be better?

Line 86 – there is one pheromone and three pheromone components.

Line 95 – I do not see this citation (29) as a good supporting reference for this statement.

Line 99 – I think it would have been more meaningful if the variation reflected intraspecific variation rather than variation among congener a novels.

We have replaced originating by "facilitating" (L. 52), added "components" to that sentence (L. 105), and replaced the citation (L. 100).

Intraspecific variation in pheromone blends is actually present within *B. anynana* and is as dramatic as variation between species. There is a population of *B. anynana* from Malawi that completely lacks MSP1 and MSP3 (Bacquet et al, Ecol. Evol. 2016). Our manipulations are comparable to this drastic natural intraspecific variation in blend. Smaller changes in blend between other putative *B. anynana* populations would require us to be able to manipulate precisely the amount of each of the three components, which can not be done since only MSP2 is available for purchase.

Overall the authors have proposed a novel series of mechanisms that would in their view facilitate the rapid evolution of new pheromone communication systems (at least in butterflies that use a strongly pheromone-biased system). It is incumbent of them to provide definitive proof of the feasibility of their model. For the reasons above, publication ought to await further confirmatory experiments that support their proposed step-wise mechanisms and correct the bioassay's deficiencies. Furthermore, the authors should recognize that although their tests achieved statistical significance, a change in the behavior of a relative small number of butterflies would have eliminated any significance by chi-squared tests.

We respectfully disagree with this comment. We are an experimental lab, and it is incumbent upon us to show, experimentally, what we have shown. Labs with modeling experience can now use our results to test whether or not the mechanism we outlined can affect evolutionary outcomes.

In our opinion, the identification of the inheritance mechanism will provide the strongest evidence for our claims, again at the experimental level. However, this will require years of additional investigation. Here we provide the first step in an ongoing research program that aims to discover and identify the mechanisms of such inheritance. We report a novel behavioral assay (3 minute odor exposure using hand-held individuals) and choice assays, that will become the basic assays in all subsequent investigations of molecular mechanism. Most importantly, we provide statistically significant behavioral results, without which we would have no basis to propel this research forward. It is, thus, important for these results to be published first.

The study of epigenetic mechanisms in insects remains incipient and is still poorly understood. Changes in DNA expression of odor receptor genes, after an odor exposure, are the only mechanisms identified so far in insects. In mice, however, changes in DNA methylation of these same odor receptor genes have been associated with the inheritance of an odor avoidance behavior. To push our project forward, there is a need to identify the nature of epigenetic mechanisms in insects, and more particularly in Lepidoptera. This investigation, however, will require multiple years of dedicated funding.

To strengthen our claims, however, we have added a novel behavioral experiment to our manuscript. We have repeated the exposure of females to NB2 and Wt2 males, and submitted their offspring to a mating assay, similarly to experiment 1. Naïve offspring of NB2- and of Wt2 -exposed females both show no mating biases for any blend. This response is similar to that of their exposed mothers, but is significantly different from the choice of untreated naïve females. We have added this information to the manuscript, and edited the respective parts accordingly (e.g. see L. 175-183, L. 280-293, Fig. 3b...).

Our sample sizes are reasonably large and we have used a conservative statistical test, so we disagree that changes in the behavior of a few individuals would have upended our conclusions. As mentioned above, we are currently pursuing the molecular mechanisms of odor learning and inheritance in

butterflies, and by performing this research we hope to ultimately validate and understand the results of the experiments described here.

Reviewers' Comments:

Reviewer #1:

Remarks to the Author:

The authors have addressed all comments and questions clearly, and I find their findings exciting and worth publishing in Nature Communications.

Reviewer #2:

Remarks to the Author:

COMMENTS TO THE AUTHORS

I like that you have added a second experiment, and in general I found the manuscript well written and interesting. However, I am still not convinced by the transgenerational effects in this research.

Lines 95-96. This sentence is unclear and needs rewording.

Lines 111-119. In this section the you need to explain the differences between the "experiment 1" and "experiment 2" you detail later.

Line 134. "innate females" is confusing

Lines 139-140. This does not clarify why it was necessary to use mixed models. It would be better to explain here that the mixed models were necessary in order to include family as a random effect – it is also confusing as the results you report in the next paragraph are chi-square test results instead. It needs to be better clarified that you used both types of tests and why you did so.

Lines 172-174. I understand your point here that it is a small effect size. But you are making a large claim here – and it seems to me that in order to demonstrate this effect you need to show that these groups are different from each other the way that you do in the maternal generation. In your responses to my comments you say: "As suggested we have compared the preference (mating bias) of naïve females from the parental generation to the preference (mating bias) of their offspring. In experiment1, naïve females form the parental generation and offspring of Wt1-exposed females prefer the Wt blend, while offspring of NB1-exposed females don't show any preference" but I do not see these statistics in the manuscript. Looking at the data my guess would be that none of the offspring generation preferences would be significantly different from the naïve maternal generation preferences.

Line 182: should be "from" not "form"

Line 395: "About 5" can you instead say the range? 3-7?

Here we respond to the reviewers' comments. We re-type the comments in black font and our response below it in blue font.

Reviewers' comments:

Reviewer #1 (Remarks to the Author):

The authors have addressed all comments and questions clearly, and I find their findings exciting and worth publishing in Nature Communications.

Thank you for your comments.

Reviewer #2 (Remarks to the Author):

COMMENTS TO THE AUTHORS

I like that you have added a second experiment, and in general I found the manuscript well written and interesting. However, I am still not convinced by the transgenerational effects in this research.

Thanks for your comments and suggestions. Please, find our detailed response to each of your points below.

Lines 95-96. This sentence is unclear and needs rewording.

We have now edited this sentence (L. 96).

Lines 111-119. In this section you need to explain the differences between the "experiment 1" and "experiment 2" you detail later.

Yes, we have now described Experiment 1 and 2 in this section (L. 117-119)

Line 134. "innate females" is confusing

We have changed "innate" to "naïve" (L 136).

Lines 139-140. This does not clarify why it was necessary to use mixed models. It would be better to explain here that the mixed models were necessary in order to include family as a random effect – it is also confusing as the results you report in the next paragraph are chi-square test results instead. It needs to be better clarified that you used both types of tests and why you did so.

We now explain why we used a GLMM - to test for the effect of the treatments, and of the male age on female mating outcomes, while allowing us to include family as a random effect factor. We also moved the chi-square test results into this paragraph (L. 141-145) and explained that this analysis test for mating biases – differences from random mate choice within each of the groups.

Lines 172-174. I understand your point here that it is a small effect size. But you are making a large claim here – and it seems to me that in order to demonstrate this effect you need to show that these

groups are different from each other the way that you do in the maternal generation. In your responses to my comments you say: “As suggested we have compared the preference (mating bias) of naïve females from the parental generation to the preference (mating bias) of their offspring. In experiment1, naïve females form the parental generation and offspring of Wt1-exposed females prefer the Wt blend, while offspring of NB1-exposed females don’t show any preference” but I do not see these statistics in the manuscript. Looking at the data my guess would be that none of the offspring generation preferences would be significantly different from the naïve maternal generation preferences.

We believe there are two ways to evaluate a change in the behavior of naïve females from the parental and the offspring generation in response to our odor exposure treatments. One way is to show, as we did, that naïve females of the parental generation had a preference for the odor of Wt males (a mating bias), whereas the naïve female offspring of NB1 or NB2-exposed mothers no longer had this preference. A second way is to show that the mating outcome of mothers (their mating choices) differs from the mating outcome of their female offspring.

The post hoc tests of the GLMM analysis compared the mating outcome of groups with each other, and the test shows that these outcomes are not significantly different from each other (the p value is above 0.05: all the statistical outcomes are indicated in Suppl. Table 1). P-values are calculated based on effect size and sample size. There is 20% more naïve offspring (of NB1-exposed mothers) that choose NB1 males compared to naïve parental females. With an effect size of 20%, it would be necessary to increase the total sample size to 275 trials in order to show that this effect size is significantly different. This sample size is too challenging because of logistical constraints, so we decided instead to perform an independent replication of this experiment using the other pheromone blend (NB2), over the course of one year. In addition, we increased the sample size (from 30 to 50 trials) to increase the precision of our estimate. This second experiment produced a similar outcome to the first experiment, providing further support that an initially strong preference for the Wt blend is lost in offspring of females once these are exposed to the new pheromone blend (NB2). The effect size observed, here again, is around 20%. As such, this second experiment would also require a much larger number of trials to show that a 20% change in mating bias is statistically significant.

However, both experiments show quite unequivocally that naïve females of the parental generation and offspring of NB1-exposed or NB2-exposed females have differences in their behavior at the level of naïve preferences. While mothers show a clear preference for Wt-blends, their offspring lose this preference and mate randomly, in both cases. This loss of preference, we believe, could be important at the population level and should be considered in models of evolution that estimate how a mutant male, with a novel pheromone odor blend, might spread its genes. When females are not able to learn a novel odor preference (or least learn to stop avoiding a novel odor), the odds of such an odor mutation spreading in a population are small, but when females can learn a new odor and subsequently alter the naïve preference of their daughters, who might also encounter similar mutant males, then the change of that mutant male spreading in the population increases, if only via neutral drift. We believe, therefore, that our experiments are relevant in the context of helping to explain the evolution of novel male pheromone blends, as well as novel preferences for those blends.

For an easier visualization of the mating bias in the different groups, we have placed them all on the same x axis and have added the corresponding 95% confidence intervals (Figures 2 and 3). We also propose in the discussion that the identification of the mechanisms that allow for this transgenerational shift in female preferences will provide the strongest evidence of our claims, but

this will require years of additional investigation since the study of epigenetic mechanisms in insects remains incipient and is still poorly understood. That's why we also have explicitly stated that understanding these results requires more experiments in the manuscript.

Finally, our lab has recently been using a very different system of odor learning and transgenerational inheritance – plant odors learned by larvae that can be transmitted to the next generation. In this system, the effect size difference between the parental generation plant odor choice and the offspring choice is ~15%, but we have fewer logistical constraints with this system and we have been able to increase our sample sizes to several hundred parental larvae and several hundred offspring. Here we have shown statistical significance in both changes in plant choice preferences across generations as well as changes in the choices in the different treatment groups (please see Gowri V. et al. (2019), *Evolution* (doi: 10.1111/evo.13861)). Evolution of odor preferences in larvae, while not being relevant for investigations of speciation via sexual selection, might become a good alternative system for investigations of molecular mechanism of odor preference learning in future.

Line 182: should be “from” not “form”

Corrected, thanks (now L. 186).

Line 395: “About 5” can you instead say the range? 3-7?

Yes, we changed it to “from 2 to 9” (L. 391).